# Exploring Sustainable Aquafeed Alternatives with a Specific Focus on the Ensilaging Technology of Fish Waste

**Anastasiia Maksimenko \*** , **Leonid Belyi, Anna Podvolotskaya** , **Oksana Son and Liudmila Tekutyeva**

Institute of Biotechnology, Bioengineering and Food Systems, Advanced Engineering School, Far Eastern Federal University, 10 Ajax Bay, Russky Island, 690922 Vladivostok, Russia; belyy_le@students.dvfu.ru (L.B.); apodvolot7777@mail.ru (A.P.); oksana_son@bk.ru (O.S.); tekuteva.la@dvfu.ru (L.T.)
\* Correspondence: maksimenko.aal@dvfu.ru

**Abstract:** The global increase in population has placed significant pressure on food security, leading to the emergence of aquaculture as a vital source of aquatic foods. However, rising costs and limited fish meal availability in aquafeeds have driven the search for alternative protein sources. While plant-based ingredients have been integrated into commercial aquafeeds, they come with challenges such as low protein content, palatability issues, and the presence of antinutritional factors. In this context, fish silage, made from fish waste and discarded fish, stands out as a promising alternative technology due to its cost-effectiveness and sustainability attributes. The production of fish silage involves the addition of organic/inorganic acids or lactic acid bacteria to homogenized fish waste, yielding a valuable mixture rich in peptides and free amino acids, offering significant nutritional benefits for animal diets. This review aims to promote sustainable practices in the aquaculture industry by analyzing research results related to ensiling technology, appraising the advantages and disadvantages of using fish silage as a feed ingredient, and focusing on emerging trends in this field.

**Keywords:** ensiling technology; fish waste; lactic acid bacteria; fish silage; waste valorization

## 1. Introduction

Aquatic foods are increasingly recognized for their significant role in ensuring food security and promoting nutrition, valued not only as rich sources of protein but also as unique and highly diverse providers of essential omega-3 fatty acids and bioavailable micronutrients [1]. By 2020, per capita consumption of aquatic food (excluding algae) had reached 20.2 kg, more than double the average of 9.9 kg per capita in the 1960s. Projections indicate that by 2030, the average consumption is expected to reach 21.4 kg per capita [2]. The surge in worldwide demand for seafood has driven rapid growth in the aquaculture sector. Global aquaculture production reached a record 122.6 million tonnes in 2020, comprising 87.5 million tonnes of aquatic animals worth USD 264.8 billion and 35.1 million tonnes of algae valued at USD 16.5 billion [2]. Predictions suggest that total aquatic animal production in aquaculture will reach 106 million tonnes by 2030 [2]. However, the aquaculture industry faces major challenges of high feeding costs and an inconsistent supply of fish meal and fish oil. Feeding expenses constitute a significant portion (approximately up to 70%) of the total operational costs in aquaculture [3–5].

Fish meal and fish oil sourced from wild-harvested fish (pelagic fish) have served as primary feed ingredients in aquaculture, providing high-quality, digestible proteins, a balanced amino acid composition, and serving as the main source of omega-3 fatty acids (specifically eicosapentaenoic acid (EPA) and docosahexaenoic acid (DHA)) [6]. However, the technology of fish meal production involves a multi-step process that demands a significant amount of energy and a continuous supply of fresh raw materials to remain economically viable [7]. In 2020, approximately 86% of fish meal was used in aquaculture, 9% was allocated for pig farming, and 5% was used for other purposes (primarily pet food

and poultry farming). Similarly, about 73% of fish oil in the same year was intended for aquaculture, 16% for human consumption, and 11% for other purposes [2]. Nevertheless, a clear trend is emerging towards reducing the presence of fish meal and fish oil in combined feed for aquaculture. This trend is primarily driven by supply fluctuations, price variations, and the continuously increasing demand from the aquafeed industry [2].

Aquaculture plays a pivotal role in global food production, but its sustainability depends on finding alternative protein sources to replace fish meal. Therefore, it is imperative to find alternative and cost-effective ingredients while reducing dependence on fish meal and fish oil for the environmental and economic sustainability of aquaculture [8,9]. Plant-based raw materials have undergone thorough investigation and are successfully included in commercial aquaculture feeds. However, utilizing conventional plant-based protein in aquaculture, particularly for carnivorous fish species, faces challenges including inadequate protein content, palatability issues, unbalanced amino acid profiles, and antinutritional factors. These challenges can detrimentally affect fish growth, feed utilization, digestibility, and overall health [10,11]. Furthermore, the use of terrestrial crops in aquafeeds, which directly competes with human food resources, introduces significant sustainability implications. These implications encompass concerns related to freshwater access, deforestation, habitat modification, ecological footprint, and potential contribution to aquatic pollution [12,13].

Besides plant-based ingredients, insect meals hold significant potential to provide the necessary protein for aquafeeds [9,14–16]. Additionally, food waste and food loss also sustain potential as valuable feed sources for aquaculture [17,18]. Instead of using them directly as feed ingredients, alternative methods like bioconversion and biotransformation of raw waste materials can be employed to enhance their utility in aquaculture feed production [9]. Bioconversion involves utilizing food waste as a nutrient source for insects (such as black soldier flies, mealworms, or crickets) and algae, which can then be used as a feed resource [19,20]. Biotransformation employs food waste as a nutrient source for microorganisms through solid-state fermentation, thereby increasing the crude protein content and reducing the fiber content of the waste. Microorganisms break down the waste, converting it into microbial biomass, which serves as a protein-rich ingredient in aquafeed [19]. Both bioconversion and biotransformation offer sustainable approaches to utilize food waste and food loss while producing valuable feed resources for aquaculture. These approaches effectively address concerns related to food waste management while simultaneously generating nutrient-rich biomass that caters to the nutritional requirements of aquaculture species [9].

Animal-based protein sources are a suitable alternative to fish meal, offering high protein content, digestibility, and lacking antinutritional components. Among commercially available animal protein alternatives, poultry by-product meal stands out for its excellent nutritional value and amino acid balance, although it may have lower levels of certain essential amino acids like lysine and methionine compared to fish meal [21–23]. The fermentation process has been used to reduce the presence of antinutritional factors in plant-based protein sources and potentially increase the nutritional quality of both animal-based and plant-based protein sources for use in aquaculture feeds [24]. By treating protein sources with appropriate microorganisms, their nutrients can be preserved and subsequently incorporated into aquafeed, potentially resulting in reduced feed costs and environmental pollution. Fermented protein meals have shown improved nutrient efficiency and the potential to enhance the nutritional value of aquafeed [11,25].

Waste generated from fish and aquaculture processing industries serves as a rich source of protein, lipids, and minerals. Ensiling has recently gained significant attention as an environmentally friendly and economically efficient technology, although it is not a new method [26]. This technology involves preserving and fermenting fish waste, including heads, viscera, skin, bones, and scales, to produce a valuable and sustainable co-product known as fish silage. Fish silage production is a simple and inexpensive alternative to traditional fish meal production, especially for small-scale processing units where investing

in a fish meal plant is not economically viable. Fish silage is a protein-rich hydrolysate with high levels of essential amino acids and can serve as a cost-effective alternative to traditional fish meal in animal diets [7,27].

Ensiling technology offers several advantages over fish meal production. The main advantage is that in areas without a fish meal production plant, fish waste can be utilized instead of being discarded, thereby reducing environmental pollution. Thus, the ensiling process is likely to be successful in areas where fresh fish waste is regularly available, and the cost of transporting it to the nearest fish meal production plant is excessively high. On the other hand, since fish processing waste quickly spoils, ensiling technology, which is easy to use and does not require expensive equipment, allows for its immediate preservation to maintain the quality of raw materials for future production of feed ingredients. Ensiling technology allows for the ensiling of any amount of raw materials, and ensiling tanks can be placed in various locations, such as on fishing vessels where fish waste is generated. Additionally, the ensiling process is milder and requires less heat, which can be advantageous in avoiding negative side reactions such as protein cross-linking [7,28].

The nutritional composition and quality of fish silage, particularly protein content, amino acid composition, and digestibility, are important factors in determining its usefulness as an animal feed ingredient [27]. The protein content and amino acid composition contribute to the high nutrient value of fish silage [29,30]. Fish silage contains essential amino acids in a balanced ratio, making it a promising protein supplement for livestock and aquaculture. The freshness and composition of the raw materials used in fish silage production are critical for the quality of the final product [29,30]. Using fresh fish waste is essential to ensure a high-quality protein source for animal feeding. Processing parameters such as the type of acid used, pH levels, additives, storage time, and temperature can affect the nutritional quality and composition of fish silage. Proper storage conditions, including time and temperature, are necessary to maintain the nutritional properties of the fish silage and prevent spoilage [27].

The utilization of fish silage in animal nutrition as a valuable feed ingredient has gained prominence due to its high nutritional value and appropriate microbiological and chemical quality [31–36]. Fish silage has found successful applications as an ingredient in animal feeds in various livestock and poultry species [37–41], including pigs [42], quails [43,44], and lambs [45]. Due to its relatively low acidity, fish silage can be directly fed without any prior mixing or treatment. This approach has been successfully employed by including fish silage as part of the daily pig feed, resulting in higher growth rates, improved health, and reduced mortality. Fish silage can also be mixed with other feed ingredients, such as grains or other dry feeds. After incorporating the fish silage, the mixture can be directly fed to livestock as a wet feed, retaining all the nutritional and health benefits for animals [27]. Furthermore, fish silage is recommended for partially substituting fish meal in animal feeds. Its hydrolyzed proteins contain abundant free amino acids and peptides, enhancing the growth performance of animals. Incorporating fish silage into extruded feeds is an effective and well-established practice, allowing it to replace a portion of fish meal (typically 5–15%) and some added water in the extrusion process. Additionally, the inclusion of fish silage has been shown to enhance the strength and durability of pellets produced by extrusion, reducing waste such as dust during transport and feeding [27,46,47]. On the other hand, fish silage can be a valuable fertilizer, especially if it does not meet the quality standards for use as animal feed. The nutrient-rich composition of fish silage makes it a suitable alternative to conventional fertilizers, with the potential to reduce environmental waste and pollution [27,48,49].

Since fish processing co-products can be classified as category 3 co-products, which are fit for human consumption, fish co-product silage can be utilized not only in animal feed but also in food applications. Fish silage can be incorporated into food products similarly to how fish protein hydrolysates are used, such as fortifying drinks, soups, sauces, dietary supplements, and sports nutrition products, whether in a liquid, semi-dried, or dried state [28,50,51]. Özyurt et al. [36] investigated the lipid quality and fatty acid compositions

of fish oils recovered from acidified and fermented fish silages. They concluded that fish oils recovered from fermented fish silages can be used not only as feed ingredients or additives for animals but also for human diets.

The objective of this review is to provide valuable insights into the evolving landscape of aquafeed ingredients, addressing sustainability challenges within the industry and presenting practical solutions. Through an examination of the latest developments, this review aims to contribute to the advancement of sustainable and cost-effective aquaculture practices. Specifically, it underscores the significance of identifying alternative and cost-effective ingredients while reducing the dependence on fishmeal and fish oil to ensure both environmental and economic sustainability in aquaculture. The review primarily focuses on ensiling technology, including acidified and fermented fish silage production, and explores its applications in aquaculture nutrition. Additionally, it describes the advantages and challenges of utilizing fish silage and innovative approaches to sustainable alternatives through waste valorization. In summary, this review aims to advance sustainable and cost-effective aquaculture practices in response to the industry's urgent demands.

## 2. Ensiling Technology of Fish Waste: A Brief Overview

The term "fish waste" generally refers to any material or parts of a fish that are discarded or not used for the primary purpose of human consumption or processing. Fish waste can include heads, tails, scales, bones, viscera (internal organs), skin, and other parts of the fish that are typically not consumed directly by humans. Additionally, fish waste may encompass fish that are caught but are not suitable for sale due to factors like size, quality, or species. More than 70% of the total fish caught undergo further processing before being introduced to the market, resulting in the generation of significant quantities (approximately 20–80%) of fish waste. The amount of waste produced varies depending on the level of processing, such as gutting, scaling, and fileting, and the species of fish, including composition, size, shape, and intrinsic chemistry. The residues from fish processing operations include muscle trimmings (15–20%), skin and fins (1–3%), bones (9–15%), heads (9–12%), viscera (12–18%), and scales (5%) [52]. Industrial fish waste and discarded fish, when not repurposed for other uses, create significant environmental problems and incur additional costs associated with landfilling or composting [53].

Fish silage production is an efficient biotechnological solution for repurposing fish waste and discarded fish from the fish processing industry. It is a low-energy, low-labor, and low-investment process that converts fish waste into a valuable product [7,27,31]. Through the ensiling process, fish waste is transformed into a liquid mixture rich in hydrolyzed proteins, lipids, vitamins, minerals, and other nutrients. These nutrients are easily digestible and absorbed by both terrestrial and aquatic animals. The liquefaction of the product results from the action of proteolytic enzymes naturally present in fish, which become available when the original raw material is milled and homogenized. Grinding the fish waste into particles smaller than 1 mm is crucial for preserving it properly, allowing for acid penetration into all cells and preventing decay in the inner parts of the particles [27].

There are two primary methods for producing fish silage, as presented in Figure 1. One method is known as acidified fish silage, achieved by adding mineral or organic acids under anaerobic conditions to lower the pH, thus preventing the growth of spoilage microorganisms and preserving the fish silage. The other method involves fermented fish silage, created through anaerobic microbial fermentation, typically using lactic acid bacteria (LAB) and added carbohydrates. Another method involves the addition of enzymes, but this method is not discussed in this review.

Fish silage production requires reasonable technological skills, and its success depends on optimizing and controlling various operating and processing, as well as environmental conditions. These factors include the fish species used as raw material, the type, strength, and amounts of acids employed, the selection of microbial strains and combinations as starter cultures, initial cell density, particle size of raw materials, the aerobic-to-anaerobic

cycle, the efficiency and maintenance of anaerobic conditions, gas evacuation in the silos, temperature, pH, buffer capacity, and moisture, among others [54].

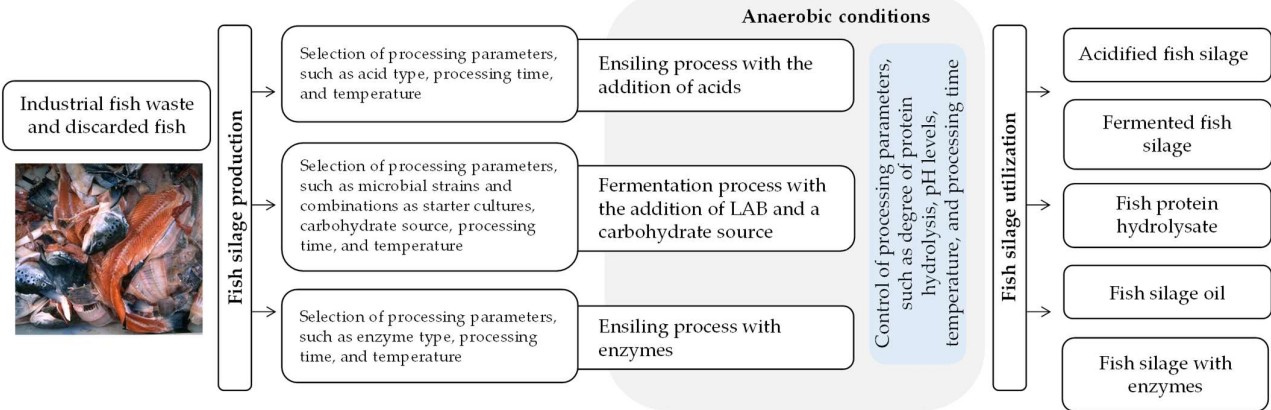

**Figure 1.** Methods for producing acidified and fermented fish silage and their utilization as alternative feed ingredients.

The type and quality of the raw materials used can influence the quality and nutritional composition of the resulting fish silage. Therefore, to produce high-quality fish silage, high-quality raw materials are required. Nutritional composition, such as protein, fat, and ash content, also varies depending on the raw material used to produce fish silage [29,30].

An acidic environment with a pH ranging from 3.5 to 4.0 is ideal for enzymatic degradation. The optimal ensiling temperature ranges from 5 to 40 °C. The choice of ensiling temperature will affect the time required for protein hydrolysis, with the process taking from several days to several weeks. Lower temperatures slow down the ensiling process, while excessively high temperatures deactivate the enzymes [27]. Fish silage can have a range of consistencies, from liquid to pasty, depending on the degree of dissolution that occurs during the ensiling process.

Fish silage production is not only a method of fish waste preservation but also the production of a mixture rich in hydrolyzed fish protein and micronutrients. Enzymes, primarily from the fish's digestive system, as well as enzymes from the skin and muscles that are active in an acidic pH range, break down proteins through autolysis into shorter peptide fractions, resulting in a liquid solution rich in low-molecular-weight nutrients. This makes the nutrients contained in the fish silage highly bioavailable and easily digestible by animals that receive it as feed [7,27].

Because short-chain peptides and free amino acids can be absorbed more easily than intact proteins and do not require prior digestion by pancreatic proteases, fish silage may be more digestible than fish meal, as suggested by several studies [32,42]. Santana et al. [32] reported that fish viscera can be converted into both acidified and fermented silage, which is a well-digested energy ingredient for aquafeed due to the high fat content (around 60%) in dry matter. The fermented silages produced with different sources of carbohydrates, such as molasses, wheat bran, and cassava waste, had a high protein content of 12% in terms of dry matter and were rich in essential amino acids, EPA, and DHA, along with beneficial microorganisms. Notably, the fermented wheat bran silage displayed the highest apparent digestibility coefficient for protein at 92%, while all formulations demonstrated an apparent digestibility coefficient for gross energy exceeding 82% [32].

However, fish silage contains a significant amount of free amino acids that can serve as precursors for biogenic amines. The low pH and specific conditions in fish silage favor the action of enzymes called amino acid decarboxylases, which are found in many microorganisms. High levels of amines can be found in fermented foods derived from raw materials with high protein content. The formation of biogenic amines in fermented foods depends on the availability of free amino acids (precursors), the presence of microorgan-

isms with amino acid decarboxylase enzymes, and suitable conditions for their growth and decarboxylating activity [33–35,55]. Biogenic amines can indeed pose a potential risk in fish products and silage, as they can be toxic to animals, leading to liver damage and reduced animal performance. Therefore, it is crucial to ensure that LAB strains used as starters in the fermentation process do not produce biogenic amines. It has been reported that the most common biogenic amines in fish and fish products are putrescine, 2-phenylethylamine, cadaverine, tyramine, histamine, spermidine, and spermine [56–60]. In a study by Dapkevicius et al. [26], LAB cultures isolated from fermented fish pastes were examined for their ability to produce histamine, tyramine, cadaverine, and putrescine. It was found that some LAB strains, particularly *Lactobacillus sakei* and *Lactobacillus curvatus*, produced these biogenic amines. Therefore, caution should be exercised when selecting bacterial strains for fish silage production. Özyurt et al. [33] investigated the chemical, microbiological, and nutritional properties of fish silages produced through acidification (3% formic acid and combination of 1.5% formic and 1.5% sulfuric acid) and fermentation (*Lactobacillus plantarum* and *Streptococcus thermophilus*). They concluded that acidified or fermented fish silage could be a valuable component in animal feeds due to its high nutritional value and appropriate microbiological and chemical quality [33]. Özyurt et al. [34] investigated the impacts of the fermentation process with acid and lactic acid bacteria strains (*Lactobacillus plantarum*, *Lactobacillus brevis*, *Pediococcus acidilactici*, *Enterococcus gallinarum*, and *Streptococcus* spp.) on the formation of biogenic amines in wet and spray-dried fish silage. The results indicated the potential use of fermented fish silage as a protein source and possibly as a probiotic ingredient in animal feed in both wet and dry forms [34].

On the other hand, fish waste contains crude lipids that are highly susceptible to oxidation due to their polyunsaturated fatty acid content and the presence of blood-derived hemoglobin. In the production of fish silage, the addition of acids can accelerate lipid oxidation, especially as pH levels decrease towards the acidic range used in ensilaging. Low pH values promote oxidation and iron ions become more available as initiators of oxidation. The weaklyacidic pH converts hemoglobin from its native oxyhemoglobin to the methemoglobin state, facilitating the release of heme groups. These heme groups can then initiate lipid oxidation through heme-mediated peroxide cleavage [28,33].

The adverse effects of lipid oxidation are well known and include a reduction in sensory quality, diminished nutritional value, and, in the case of feed applications, lower feed conversion ratios and reduced carcass quality, as observed in fish and broilers [28,61]. To prevent lipid oxidation and stabilize lipids in animal feeds, the addition of antioxidants such as ethoxyquin is crucial, allowing for extended storage [7]. Additionally, butylated hydroxytoluene (BHT) is an effective antioxidant when included at a dosage of 100–250 mg/kg, effectively inhibiting lipid oxidation in various types of fish silages during storage. This ensures that lipid oxidation remains below the threshold value of 8 mg of malonaldehyde per kilogram (kg) [33].

Yano et al. [62] conducted an interesting study on reducing lipid content in fish meal prepared from fish waste using *Yarrowia lipolytica* yeast, as a high lipid content in fish waste can diminish its quality. During the study, microorganisms capable of recovering raw lipids from fish mince samples through solid-state fermentation were examined. A strain of *Yarrowia lipolytica* was identified, demonstrating the highest efficiency by reducing lipids by 29%. The effectiveness of lipid reduction was significantly influenced by the ratio of surface area to the weight of the fermented mince samples and water content, emphasizing the importance of oxygen supply. The fermentation process over a 96 h incubation period led to an efficiency of reducing crude lipids reaching 46%, indicating an increase in protein content in the final product. Additionally, fermentation suppressed the carbonyl value, a measure of lipid oxidation. These results indicate that fermentation has the potential to improve the quality of fish meal obtained from lipid-rich fish waste [62].

## 2.1. Acidified Fish Silage Production

Preserving fish processing waste through acid preservation is a simple and cost-effective method suitable for various scales of operation [7,63,64]. In traditional acidified fish silage production, organic or inorganic acids, or a combination of both, are added at around 2–3% ($w/w$) to homogenized fish waste to lower the pH to 4 or below, effectively preventing the proliferation of pathogenic and spoilage microorganisms. The choice of acid type depends on cost and availability. Organic acids like formic acid, although more expensive than mineral acids, produce less acidic fish silages that do not require neutralization before use [27,63]. Once the fish waste is acidified, temperature-dependent autolytic liquefaction takes place due to the action of endogenous proteolytic enzymes, primarily pepsins found in the viscera. Autolysis occurs at a slower rate in fish waste without stomach-containing viscera, unless acid proteases are added [7,61]. Proteins in acidified fish silage are largely hydrolyzed into free amino acids and short-chain peptides, and lipids into fatty acids.

The use of organic acids in feed serves as both a simple method to preserve fish processing waste and a potential enhancer of the well-being and growth of farmed animals and fish. Short-chain organic acids like formic acid are being explored as growth promoters in animal feed, particularly in poultry and pig diets, as alternatives to banned non-therapeutic antibiotics [7]. These organic acids exert antimicrobial effects, primarily in the upper part of the gastrointestinal tract, and can influence the microbiota in both the feed and the gut. Additionally, some organic acids possess antioxidant properties, contributing to feed safety and preservation [54]. Moreover, these organic acids may improve the absorption of specific minerals such as calcium and phosphorus, with potential benefits for animal health and growth [7].

Formic acid in the fish silage hydrolysate can promote the growth and well-being of fish, especially in unfavorable microbiological conditions. This may encourage fish processing plants to preserve fish waste using silage and feed manufacturers to include these products in feed [7]. However, the effectiveness of short-chain organic acids in aquaculture feed can vary depending on factors such as the type of acid used, the form in which they are applied (e.g., salts), the buffering capacity of the feed, and environmental microbial conditions. There are limitations on the amount of formic acid that can be used in fish feed without negatively impacting growth and health. A recent in vitro study suggested that the utilization of free amino acids and short-chain peptides derived from salmon by-product acidified silage could potentially enhance the health and welfare of farmed fish during stressful periods [65].

## 2.2. Fermented Fish Silage Production

Fermented fish silage is produced by adding microorganisms from the LAB group, such as *Lactobacillus* sp. and *Lactococcus* sp., along with a carbohydrate source such as molasses or fruit/vegetable processing waste. The LAB are the primary microorganisms used to ferment the fish waste to produce the fermented fish silage. During the fermentation process, under anaerobic conditions, the LAB ferment carbohydrate sources to produce lactic acid, antibacterial compounds (bacteriocins), and energy. This process leads to a decrease in pH to values around 3.5–4.0 in the fish silage. These compounds promote silage conservation by inhibiting the growth of pathogenic microorganisms and by allowing the action of the proteolytic enzymes present in fish viscera, which collaborate in the hydrolysis and dissociation of fish waste [7,32,43,66,67]. High-quality silage is obtained when lactic acid becomes the dominant component due to its antibacterial properties and its ability to rapidly lower the pH of the fish silage [54].

The selection of LAB strains is crucial for the successful production of fermented fish silage. Studies have indicated that using LAB isolated from the same source is the most effective approach for fermentation, as reported by Rai et al. [68] and Özyurt et al. [33]. Thus, the production of fermented fish silage holds significant potential for converting underutilized protein into a valuable hydrolyzed product. However, it is important to note

that this method is more complex and requires careful monitoring throughout the entire fermentation process.

The fermentation process provides protection for the lipid and protein content of fish silage, making it a more suitable option for animal feed [68]. It has been reported that using LAB for the fermentation of fish waste can lead to the recovery of various biomolecules and enhance the stability of fat in fish silage. Lactic acid fermentation improves the stability of fat in fish silage by reducing lipid oxidation, making it a better option for animal feed. Furthermore, LAB are capable of producing compounds like bacteriocins, diacetyl, hydrogen peroxide, and organic acids, all of which play a role in preventing fat oxidation in fish silages [31]. In addition to these benefits, some LAB have the capability to degrade biogenic amines using amino oxidases [26]. However, it should be noted that oil extracted from fermented fish silage contains higher levels of free fatty acids, which could potentially limit its suitability for use in feed [33,68].

The fermentation process significantly enhances the nutritive value of fish silage. Numerous studies have documented that the utilization of the fermentation process leads to an increase in the crude protein content and the LAB count in fish silage. Furthermore, incorporating fish silage into animal diets has been shown to improve both growth performance and the intestinal health of animals, making it a suitable alternative protein source to fish meal or soybean meal in animal diets [37–39]. Shabani et al. [37] reported that the fermentation process resulted in an increase in the crude protein content (from 570.2–608.1 g/kg) and LAB count in fermented fish silage. Nonetheless, Mach and Nortvedt [69] and Özyurt et al. [34] reported that the production of acidified or fermented silages resulted in minor variations in crude protein and lipid contents compared to the raw materials.

Tropea et al. [66] demonstrated an effective approach to utilizing fermented fish and lemon peel waste as a value-added product for aquafeed, offering a protein-rich alternative (up to 48.55%) to conventional protein ingredients and contributing to healthier and more sustainable aquafeeds. The authors suggested that the capability of the starter cultures to grow at low pH can be ascribed to the supplementation of lemon peel during the fermentation process. Polysaccharides, such as pectins, show a protective effect on LAB against low pH. Their ability to achieve this in fermenting fish waste supplemented by lemon peel was confirmed by the increasing protein level during the process, making these wastes an excellent raw material for aquafeed production with *Lactobacillus reuteri* and *Saccharomyces cerevisiae* [66].

Inoue et al. [70] fermented non-sterilized fish waste with a combination of starter cultures, including film-forming yeast (*Candida ethanolica*) and LAB (*Lactobacillus casei* and *Lactobacillus rhamnosus*), to create a liquid broth rich in amino acids and minerals. The fermentation process maintained a high level of beneficial bacteria and suppressed the growth of harmful bacteria. The fermented broth contained all 20 $\alpha$-amino acids, which make up proteins, and various minerals in abundance. Mice fed with the fermented broth showed a significant weight increase compared to those without it, and their blood analysis indicated improved immune cell percentages without any observed abnormalities in organs or behavior [70].

It has been reported that proteins from fermented fish silage are more digestible than those from acidified fish silage due to the autolytic action catalyzed by the enzymes present in fish waste that degrade proteins into short peptides and free amino acids [71]. Ramírez et al. [43] reported that the protein hydrolysis occurring during fermentation increased the digestibility of fish silage from 69% to 81.6%. This improvement in digestibility is attributed to the release of peptides and free amino acids resulting from protein hydrolysis, which are considered potential chemo-attractants and nutritious stimulants in carnivorous and other animal species [72].

### 2.3. Fish Silage Oil Production

The oil extracted from fish silage can serve as a valuable feed ingredient, particularly suitable for aquaculture feed [73–75]. Several studies have illustrated the effective utiliza-

tion of the ensiling process to recover oil from fish waste. Fermentation has been shown to successfully recover over 85% of the oil from fish viscera [68]. In a study conducted by Raeesi et al. [73], a comparison of the qualitative properties of silages prepared from fish viscera and the oils extracted from them was undertaken. Fish viscera silages were prepared using both acidic and fermentation processes and were stored at room temperature for 30 days. The results indicated that fermented fish viscera silage exhibited a greater potential for producing extracted oil, while acidified fish viscera silage could serve as a protein source in animal feed [73].

Özyurt et al. [36] conducted an investigation into the lipid quality and fatty acid compositions of fish oils obtained from fish waste silages produced with formic acid and five different LAB strains, including *Lactobacillus plantarum*, *Pediococcus acidilactici*, *Enterococcus gallinarum*, *Lactobacillus brevis*, and *Streptococcus* spp. Their findings suggested that the initial lipid quality of fermented fish silages was superior to that of acid-preserved fish silage. Furthermore, they concluded that fish oils recovered from fermented fish silages have the potential to serve as feed and food additives or supplements for both animal and human diets [36].

Fish silage oil, derived from fish processing waste, holds promise as a fish feed ingredient due to its potential as a rich source of essential fatty acids, particularly polyunsaturated fatty acids (PUFAs), and as a cost-effective substitute for conventional fish oil in aquaculture. Moreover, it presents the advantage of being a more sustainable choice when contrasted with traditional fish oil, which relies on wild-caught fish. Incorporating fish silage oil into fish feeds not only provides essential nutrients but also contributes to promoting sustainability within the aquaculture sector [74–76].

## 3. Nutritional and Health Benefits of Fish Silage

Fish silage is a versatile product derived from various raw materials, including whole fish, fish waste, or a mixture of different fish parts. The nutritional composition and quality of fish silage depend significantly on the type and freshness of the raw materials used. Parameters such as protein, fat, and ash content can vary depending on the raw materials employed in the production process [7,27,29,30]. Typically, fat-free fish silage contains approximately 80% moisture, 15% protein, and less than 4% ash [27]. The condition and quality of raw materials play a crucial role in determining the final quality and nutritional profile of the fish silage [29,30,32]. van 't Land et al. [30] conducted research on the effect of raw material composition on the nutritional quality and stability of fish silage. They used different combinations of whole undersized flatfish and codfish to produce four fish silages using 2.5% formic acid. They concluded that the nutritional quality of fish silage strongly depends on the freshness and composition of the raw materials. The raw material combination affects the nutritional quality and stability of fish silage. Deamination after extended autolysis leads to protein and amino acid losses, while oxidation processes cause a decrease in lipid quality [30]. Fish silage shares nutritional qualities with fish meal but offers improved digestibility due to hydrolyzed proteins and lipids [27,29,32]. The nutritional value of fish silage lies in its high protein digestibility, attributed to the high degree of protein hydrolysis and the presence of essential amino acids. The degree of protein hydrolysis should be considered a key chemical quality criterion for fish silage, as autolysis (excessive hydrolysis) and rancidification (lipid oxidation) can negatively impact the nutritional quality and stability of the product [29,30]. The chemical composition, quality, and nutritive value of fish silage can vary markedly depending on the source and freshness of the raw materials. Using a diverse mixture of raw fish materials may help improve the amino acid and fatty acid profiles of the resulting fish silage.

Banze et al. [29] compared the composition and quality of acidified fish silage produced from tuna viscera and assessed its nutrient digestibility compared to fish meal. The fermentation period was 30 days, during which 61.74% of the crude protein was solubilized. The acidified fish viscera silage exhibited good nutritional composition, including high protein content, a favorable amino acid profile, and essential fatty acids, and showed good

microbiological quality. Crude protein digestibility was similar between fish viscera silage and fish meal at 88.52%, but dry matter digestibility was higher for fish viscera silage at 92.20%. The authors concluded that fish viscera silage is of high nutritional quality and has good nutrient digestibility, indicating its potential as an alternative protein source in aquafeeds [29].

### 3.1. Protein Content of Fish Silage

In the research by Shabani et al. [37], an increase in the crude protein content from 570.2 to 608.1 g/kg was observed in fermented fish silage by *Lactobacillus plantarum* and *Aspergillus oryzae*. In the study by Banze et al. [29], the crude protein content was 669.40 g/kg dry matter in acidified fish viscera silage after 30 days of ensiling. Tropea et al. [66] reported that the microorganisms were able to feed on fermenting fish waste with the addition of lemon peel, increasing the protein content during the process to 48.55%, making this waste an excellent raw material for the production of aquafeeds with *Lactobacillus reuteri* and *Saccharomyces cerevisiae*. The study by van 't Land et al. [30] showed that the crude protein content can decrease during the ensiling process. After 91 days, the protein content in four types of acidified fish silage decreased from 739 to 580 g/kg dry matter. The authors suggested that the decrease in protein content could be due to a higher degree of protein hydrolysis, which reached about 60–62% after 91 days. Extended protein hydrolysis leads to overall deamination, which also results in a reduction in essential amino acids and an increase in total volatile basic nitrogen [30].

### 3.2. Amino Acid Profile of Fish Silage

Banze et al. [29] compared the amino acid profiles of acidified fish viscera silage and fish meal and concluded that the silage presented a profile of essential and non-essential amino acids numerically superior to that of fish meal for all amino acids, except for glycine. The essential amino acid content (g/100 g) of fish viscera and fish meal, respectively, is as follows: histidine (2.11 vs. 1.24); arginine (8.04 vs. 3.71); threonine (5.02 vs. 2.29); valine (1.87 vs. 0.99); methionine (2.89 vs. 1.47); lysine (3.70 vs. 3.32); isoleucine (4.24 vs. 2.36); leucine (6.12 vs. 3.75); phenylalanine (4.08 vs. 2.55). The non-essential amino acid content is as follows: alanine (5.30 vs. 4.40); proline (3.62 vs. 3.42); tyrosine (5.75 vs. 5.05); serine (5.12 vs. 3.40); glycine (5.08 vs. 6.20); aspartic acid (7.10 vs. 5.72); glutamic acid (14.60 vs. 10.70); cystine (3.64 vs. 1.87). Santana et al. [32] reported that all essential amino acids necessary for the diet of fish were present in the produced fish silages. Specifically, in fermented fish silage, the essential amino acids present in higher proportions were leucine, arginine, and lysine. Glutamic acid was the most frequently detected non-essential amino acid in all fish silages. The high concentration of glutamic acid present in fish silage can affect the immune system of animals, promoting the synthesis of cytokines, immune-modulating substances necessary for inducing lymphocyte proliferation. Low pH values of the silage did not contribute to the stability of tryptophan, especially when it was not bound to protein, resulting in low tryptophan content in the experimental fish silages [32].

### 3.3. Lipid Content and Fatty Acids Composition of Fish Silage

The fish silage can be a source of beneficial fatty acids, including monounsaturated fatty acids (MUFAs), polyunsaturated fatty acids (PUFAs), and highly unsaturated fatty acids (HUFAs), in aquafeed formulations. MUFAs contribute to fish health when constantly ingested, promoting the reduction in oxidized low-density lipoprotein in the plasma, preventing the transport of cholesterol to tissues, and inhibiting body fat accumulation [32,76]. The fatty acid composition of the fish viscera silage showed that the content of the n−3 fatty acid series, including EPA and DHA, was 6.07% and 21.72%, respectively. The n−6 polyunsaturated fatty acids, linoleic acid, and arachidonic acid, were present at 3.19% and 3.04%, respectively [29]. The presence of HUFAs, such as EPA and DHA in fish silage, can enrich diets for omnivorous fish, which generally have a low HUFA content due to the use of plant-based or terrestrial animal by-product ingredients [32].

In the study by Santana et al. [32], the fatty acids present in the fermented viscera silages were predominantly unsaturated fatty acids. Acidified viscera silage presented the highest content of saturated fatty acids (SFAs). Molasses viscera silage showed the widest variation between the contents of SFAs and MUFAs, with 20.1% higher MUFAs versus SFAs, unlike acidified viscera silage, which contained 6.2% more SFAs than MUFAs. Oleic (18:1 n−9) and linoleic (18:2 n−6) fatty acids showed the highest concentration among MUFAs and PUFAs, respectively. Viscera silages had a lower n−6:n−3 ratio compared to tambaqui viscera [32].

Ozyurt et al. [36] studied the lipid quality and fatty acid compositions of fish oils recovered from acidified and fermented fish silages using five different LAB strains. They found that the fish oils from these silages exhibited valuable fatty acid compositions and high lipid quality. The SFA contents were 17.56–19.12% and 23.27–23.64%, respectively, while the MUFA contents were 44.31–45.36%, respectively. The most abundant fatty acid in the total PUFAs of the silages was linoleic acid (15.36–16.01%). Additionally, the contents of DHA in acidified and fermented silages were 3.59–4.31%, respectively, and EPA contents were 2.32–2.66%, respectively [36].

The fatty acid content during fermentation showed a significant increase in MUFAs and a decrease in PUFAs, while SFAs remained unaffected. The main fatty acids detected were C18:1 n−9 cis, C18:2 n−6 cis, and C16:0. Some fatty acids increased significantly, while others decreased. This composition is considered suitable for aquafeed formulations due to the extended shelf life resulting from the decrease in PUFAs [66].

Goosen et al. [74] evaluated fish waste silage oil as an alternative to commercial pelagic fish oil in tilapia diets and determined its effects on fatty acid profiles, production parameters, gut microbiota, and gut histology. Fish silage oil successfully replaced commercial oil without negatively affecting production performance and also improved cellular non-specific immunity by 33%. It turned out to be a good source of PUFAs (36.9 g/100 g of total fatty acids), exhibited antimicrobial properties in the feed and gastrointestinal tract, and caused a significant shortening of intestinal folds (34.4%) in the midgut of experimental fish. Fish silage oil is a cost-effective alternative to edible oil for tilapia diets, offering advantages over some traditional fish oils [74–76].

In the study by van 't Land et al. [30], crude lipid decreased in acidified fish silage or remained stable after 91 days. The decrease in crude lipid in acidified silage could be the result of lipid oxidation, as reflected by the TBARS value and decreased PUFAs content. Overall, there is a significant effect of raw material composition on the nutritional quality and stability of fish silage. A more diverse mixture of raw materials improved nutritional quality, mainly in the form of essential amino acids and PUFAs, but was also more prone to deterioration by chemical and biological processes [30].

### 3.4. Ash Content of Fish Silage

The ash content of fish silages depends on the composition of the raw material and can range from 11.9% to 21.5% [32]. Santana et al. [32] reported that the silages had low ash content because only viscera were used in the silage formulation. Although fish viscera silage has a lower protein content compared to fish meal, fish meal contains five times more ash than fish viscera silage. High ash content is undesirable in fish diets, as some mineral needs can be met directly through the water in the aquaculture system [32]. Fish silage produced from fish viscera (intestine, stomach, liver, pancreas, swimming bladder, kidney, spleen, gonads), without bones, fins, and heads, had low mineral content [29]. This low mineral content in the fish viscera silage is considered a positive aspect, as excess minerals in fish diets are undesirable [29]. van 't Land et al. [30] reported that the ash content slightly decreased, from 192 g/kg dry matter to 176 g/kg dry matter, in four different types of acidified fish silages after being stored for 91 days at ambient temperature [30]. During the fermentation process, the ash content decreased significantly from 0.83% to 0.66%. This may be due to the partial use of ash by yeast as a source of minerals during fermentation [66].

### 3.5. Microbial Characteristics of Fish Silage

Tropea et al. [66] reported that rapid pH reduction is necessary to maintain microbial hygiene and product quality in aquafeed. Microbiological analysis to determine the total coliform count showed a general decrease during fermentation, reaching complete absence after 96 h. The reduction in coliform bacteria may be related to inhibitory compounds (bacteriocins) produced by microorganisms used during lactic acid fermentation and/or acidification of the environment. Furthermore, the reduction in coliform count can provide good bioconservation against unwanted and/or harmful microorganisms. The final fermented products contained few spoilage microorganisms and were rich in beneficial microorganisms, representing a healthy final substrate with added value [66]. According to Banze et al. [29], the microbiological analysis did not indicate significant growth of microorganisms in the raw material or in the silage produced, which shows the good quality of the fish viscera used. The absence of microorganisms in the silage also reflects the important role of acetic acid in preventing the proliferation of microorganisms during the process.

### 3.6. Biogenic Amines

Biogenic amines are low-molecular-weight decarboxylation products of amino acids formed during microbial fermentation. Several fermented foods may contain biogenic amines, such as histamine, tyramine, and/or phenylethylamine, at levels above documented toxic doses. The levels of biogenic amines can be controlled with proper selection of raw materials, storage, processing, and hygiene practices [77]. Natural antimicrobials of plant origin can be used in fish and seafood, emphasizing their mechanisms of action to prevent bacterial growth and the formation of biogenic amines [78].

During the fish silage production process studied by Banze et al. [29], the concentration of biogenic amines such as cadaverine, histamine, tyramine, agmatine, phenylethylamine, and tryptamine increased, except for putrescine and spermidine. The authors suggested that this increase may be related to the high ambient temperature (28 °C) during the ensiling process. The study by Özyurt et al. [34] investigated the impact of fermentation with acid and lactic acid bacteria strains on the formation of biogenic amines in fish silage. The predominant biogenic amines identified included cadaverine, putrescine, spermidine, spermine, serotonin, dopamine, and agmatine. The authors concluded that fermented fish silage could be a potential protein source and possibly a probiotic ingredient for animal feed, both in wet and dry forms [34].

### 3.7. Protein and Lipid Digestibility of Fish Silage

In the study by Banze et al. [29], the apparent digestion coefficient was similar between fish viscera silage and fish meal, at 88.12–88.92%. In terms of dry matter, the silage was more digestible (92.20%) than fish meal (83.84%). The authors suggested that the high dry matter digestibility of the silage, compared to that recorded for fish meal, is probably due to the acid hydrolysis [29]. Santana et al. [32] reported that fish viscera silage showed high digestibility in tambaqui juveniles. The protein hydrolysis of the silage into short-chain peptides and free amino acids makes it more bioavailable for digestion and absorption by fish. The apparent digestibility coefficient values of crude protein from fish viscera silage were similar to the apparent digestibility coefficient values of fish meal and corn gluten for tambaqui, ranging from 84.24% to 91.94% dry matter [32].

The fish viscera silages exhibited a high lipid content (above 59%), and tambaqui juveniles showed an apparent digestion coefficient for lipids above 93%. The apparent digestion coefficient of lipids from the silages was similar to that of fish oil (92%) and higher than that of the corn (85.8%) and soybean (85.1%) oils evaluated for tambaqui juveniles [32]. Fish viscera silage also showed a similar apparent digestion coefficient of gross energy to those of salmon meal (81.1%), poultry byproduct meal (83.8%), and defatted black soldier fly larvae (86.6%). Additionally, it was higher than that of feather meal (77.2%) and tilapia processing waste meal (70.2%) when evaluated for tambaqui juveniles [32].

### 3.8. Beneficial Effects of Fish Silage on Animal Health and Feed Quality

Organic acids in fish silage, as reported by Toppe et al. [27], exhibit antibacterial properties in the animal's intestine and act as natural preservatives within the fish silage. Kuley et al. [54] conducted a study on the organic acid contents of acidified and fermented fish silages. They concluded that various selected LAB strains have a good capability to produce significant amounts of organic acids, especially lactic, acetic, and propionic acids, during the 3-week fermentation of fish silages. In terms of food safety and quality, the production of relatively high amounts of organic acids in wet and spray-dried fish silages indicates their suitability for use in animal feed [54]. Furthermore, terrestrial livestock and aquaculture animals fed with organic acid-supplemented diets have demonstrated enhanced feed intake, improved growth performance, increased feed utilization effectiveness, and health-promoting effects [79–81].

Peptides and free amino acids resulting from protein hydrolysis have the potential to stimulate the non-specific immunity of aquatic animals [76,82]. Fish viscera silage has been demonstrated to stimulate the cellular non-specific immunity of fish species like *Oreochromis mossambicus*. This immune stimulation is attributed to the protein hydrolysis products present in the fish viscera silage, rather than formic acid. Furthermore, fish viscera silage serves as a valuable source of dietary protein and essential amino acids in fish diets. The level of silage inclusion in the diet can influence both fish growth performance and the enhancement of cellular immunity. Incorporating fish viscera silage with protein hydrolysis products into fish diets can have positive effects on both growth and immune response in fish [76,82].

The composition of fish feed plays a significant role in influencing the gut morphology and overall health of fish. Healthy fish typically exhibit longer folds and increased villus height in their intestines, reflecting efficient nutrient absorption and overall well-being. Conversely, fish with shorter folds and reduced villus height may experience decreased nutrient absorption, compromised immune function, and subsequently, reduced growth performance due to poor nutrient utilization. Therefore, the structural integrity of the fish's gut is crucial for its digestive capabilities [21,83,84].

There is research demonstrating that feeding fish with fermented diets results in a higher microvilli density compared to control diets. This enhancement is attributed to the fermentation effect of the feed by the probiotic bacteria *Lactobacillus casei*, resulting in a more beneficial impact compared to using non-fermented feed alone [85]. Additionally, Wang et al. [86] reported an improvement in the intestinal morphology (intestinal folds, enterocytes, and microvilli) of young turbots (*Scophthalmus maximus*) fed soybean meal fermented by *Lactobacillus plantarum*, in comparison to a non-fermented diet. A study by Davies et al. [87] demonstrated that acidified fish silage influenced gut morphology, displaying variations in mucosal fold features, while fermented fish silage affected the gut perimeter ratio. Despite lower dry matter digestibility in experimental diets, protein and energy digestibility remained at acceptable levels in diets enriched with LAB and organic acids [87]. Furthermore, fermented feeds, enriched with LAB and organic acids, enhance gut health and nutrient absorption in fish.

Góngora et al. [88] investigated biological fish silage using a selected strain of LAB (*L. arizonensis*) and chemical fish silage by adding sulfuric acid and formic acid. BALB/c mice were fed isoenergetic diets containing 36.3% (*wt/wt*) of biological and chemical fish silage, respectively, for 30 days. Both additives did not provoke lesions in the gut, thinner wall, distension, or abnormal vascularization. Mice fed biological fish silage showed a higher concentration of LAB in their gut ($2.51 \times 10^4$ cfu LAB/g vs. $3.98 \times 10^3$ cfu LAB/g), as well as a higher weight gain ($23.8 \pm 3.8$ g vs. $16.7 \pm 3.7$ g), feed conversion ratio (4.12 vs. 6.71), protein efficiency rate (0.69 vs. 0.63), and villi height (455 μm vs. 418 μm) compared to those fed chemical fish silage, indicating a potential probiotic effect of *L. arizonensis* [88].

Shabani et al. [37] studied the impact of adding increasing amounts of fermented fish silage to broiler chicken diets. Fish waste was fermented with *Lactobacillus plantarum*

and *Aspergillus oryzae*. The study found that as the inclusion of fermented fish silage increased, the feed conversion ratio improved significantly over 42 days. Liver weight and LAB population in the crop increased, while the coliform population in the ileum decreased with higher concentrations of fermented fish silage. Substituting plant-based meal with fermented fish silage also improved the villus height to crypt depth ratio in the jejunum. Overall, the fermentation process improved the nutritional value of fish waste, and feeding fermented fish silage enhanced the growth and intestinal health of broiler chickens, suggesting it could be a viable protein source to replace soybean meal in their diet [37–39].

In addition, silage oil exhibits antimicrobial properties that benefit both the quality of feed and the gastrointestinal tract of fish. This antimicrobial effect can be advantageous for producers, as it improves feed hygiene and reduces bacterial competition for nutrients in the gastrointestinal tract [74]. In terrestrial animal husbandry, reducing intestinal microbes is a common practice to enhance production efficiency by making more nutrients available to the host animal. While the specific cause of decreased microbial numbers in the feed and gastrointestinal tract of fish remains unknown, it may be related to the presence of formic acid in silage oil. Formic acid is known for its antimicrobial properties, and similar substances have been shown to alter the composition of intestinal microflora in tilapia and reduce bacterial numbers in their feces [74,75].

## 4. Utilization of Fish Silage in Aquaculture Feeds

The utilization of fish silage as a feed ingredient for various fish and crustacean species offers several advantages, including cost-effectiveness, sustainability, and the potential to improve the nutritional value of aquaculture diets [87,89]. Fish silage can be used alone as acidified or fermented silage, or as a protein hydrolysate, or in combination with plant-based ingredients in co-dried form. Several studies have reported the effective utilization of acidified or fermented fish silage, as well as protein hydrolysate, when included in moderate amounts in aquafeed (Table 1).

**Table 1.** Utilization of acidified or fermented fish silage in aquafeeds.

| Aquatic Animal | Feeding Trial | Ensiling Conditions | Results | Reference |
|---|---|---|---|---|
| Black Bass (*Micropterus salmoides*) | 66 days | Acid-treated fish silage | Up to 15% acidified fish silage can be used as a partial substitute for fish meal in the formulation of carnivorous fish feed | [90] |
| Japanese sea bass (*Lateolabrax japonicus*) | 14 days | Protein hydrolysate produced from acid-treated fish silage | Enhanced growth performance of Japanese sea bass is observed when 15% of the fish meal is replaced with silage protein hydrolysate | [91] |
| Atlantic salmon (*Salmo salar*) | 91 days | Protein hydrolysate produced from acid-treated fish silage | The best growth performance of Atlantic salmon is observed when silage protein hydrolysate is included in the diet at levels below 15% | [92] |
| Nile tilapia (*Oreochromis niloticus* L.) | 56 days | Shrimp head protein hydrolysate | Shrimp head protein hydrolysate is a promising alternative protein source for feeding tilapia, and it can improve the growth rate even at dietary inclusion levels as high as 15% | [93] |
| Orange-spotted grouper (*Epinephelus coioides*) | 42 days | Protein hydrolysate produced from acid-treated fish silage | The combination of 10% or 20% silage protein hydrolysate with poultry by-product meal could replace 50% of fish meal protein in the diets without any adverse effects on growth performance | [94] |

**Table 1.** *Cont.*

| Aquatic Animal | Feeding Trial | Ensiling Conditions | Results | Reference |
|---|---|---|---|---|
| African catfish (*Clarias gariepinus*) | 70 days | Fermented fish silage, which was produced through fermentation by *Lactobacillus plantarum* using carbohydrate substrates such as molasses, was co-dried with soybean meal, poultry by-product meal, hydrolyzed feather meal, and meat and bone meal | Fermented fish silage co-dried with protein feedstuffs is a suitable protein supplement, capable of providing up to 50% of dietary protein without adversely affecting feed efficiency, fish growth, or health | [95,96] |
| Nile tilapia (*Oreochromis niloticus*), African catfish (*Clarias gariepinus*) | 70 days | Co-dried fermented fish silage and soybean meal | Co-dried fermented fish silage and soybean meal can be used as partial replacements for fish meal protein in dry aquaculture diets | [97] |
| Catfish (*Clarias gariepinus*) | 14 days | Raw heads of river prawn were fermented with *Lactobacillus plantarum* using molasses or cassava starch as the carbohydrate source; hydrolyzed feather meal, poultry by-product meal, or soybean meal, used as an alternative filler, was blended with the liquid silage and solar-dried | Dried shrimp head silage meal is a suitable and promising protein feedstuff for fish diets; the digestibility coefficients of dry matter, crude protein, gross energy, and essential amino acids in the silage fed to catfish fingerlings exceeded 70% | [98] |
| Nile tilapia (*Oreochromis niloticus*) | 15 days | Fermentation by *Lactobacillus plantarum* using carbohydrate substrates such as molasses; the wet silage was combined with poultry by-product meal, a blend of soybean-hydrolyzed feather meal, or menhaden fishmeal for pellet production | Moist fermented fish silage pellets are both physically stable and highly digestible by Nile tilapia, making them suitable as farm-made fish feeds | [46] |
| Nile tilapia (*Oreochromis niloticus*) | 30 days | Fermentation by *Lactobacillus plantarum* using carbohydrate substrates such as molasses, corn flour, or tapioca flour | Co-dried fermented fish silage is a suitable protein feedstuff in fish diets; the pellets produced from fermented silage demonstrate higher digestibility and excellent water stability | [47] |
| Nile tilapia (*Oreochromis niloticus*) | 90 days | Dried fermented fish silage was combined with tomato by-product meal and potato by-product meal in a proportion of 30:40:30 $w/w/w$ | Replacing 30% of dietary protein with dried fermented fish silage in tilapia diets does not have adverse effects on growth or feed utilization parameters | [99] |
| Nile tilapia (*Oreochromis niloticus*), African catfish (*Clarias gariepinus*) | 90 days | Fish silage was prepared by fermenting fish waste (60%), yogurt (5%) as a source of *Lactobacillus plantarum*, molasses (5%), and rice bran (30%) as a filler for 30 days | Replacing 25% of fish meal with dried fermented fish silage in tilapia diets and 50% of fish meal in catfish diets does not significantly adversely affect the growth or feed utilization parameters of the fish | [100] |

**Table 1.** *Cont.*

| Aquatic Animal | Feeding Trial | Ensiling Conditions | Results | Reference |
|---|---|---|---|---|
| Nile tilapia (*Oreochromis niloticus*) | 84 days | Fermented fish silage was prepared by mixing fish waste (60%), rice bran (30%), dried molasses (5%), and yogurt (5%) as a source of *Lactobacillus* spp. for the lactic acid anaerobic fermentation process over 30 days | Replacing up to 50% of fish meal with dried fermented fish silage does not have any negative effects on the growth and feed utilization of tilapia; additionally, it results in a 15.59% reduction in feeding costs | [101] |
| African catfish (*Clarias gariepinus*) | 90 days | Fermented fish silage was prepared by mixing fish waste (60%), rice bran (30%), dried molasses (5%), and yogurt (5%) as a source of *Lactobacillus* spp. for the lactic acid anaerobic fermentation process over 30 days | Replacing 50% of fish meal with dried fermented fish silage in diets does not significantly adversely affect the growth or feed utilization parameters of catfish, and this replacement reduces feed costs | [102] |
| Olive flounder (*Paralichthys olivaceus*) | 70 days | A mixture of fermented fisheries by-products and soybean curd residues | Up to 30% of fish meal can be replaced by this mixture without affecting the growth performance of juvenile olive flounder | [103] |
| Catfish (*Heteropneustes fossilis*), Indian major carp (*Labeo rohita*) | 60 days | Fish offal wastes were fermented, along with mustard oil cake and rice bran, using a mixture of a commercial suspension of microorganisms, molasses, and water | Fermented fish offal can be included up to a 30% level as a partial replacement for fish meal in the formulation of the fish diet | [104,105] |
| European sea bass (*Dicentrachus labrax*) | 63 days | Apple pomace fermented fish silage, molasses fish silage, and acidified fish silage | Fish silage produced by formic acid or through fermentation with carbohydrate sources and lactic acid bacteria is an effective partial replacement for fish meal in aquaculture feeds | [87] |
| Mozambique tilapia (*Oreochromis mossambicus*) | 52 days | Fish viscera silage produced from acid ensiling | Fish viscera silage can serve as a source of dietary protein and essential amino acids in tilapia diets. The viscera silage can stimulate the cellular non-specific immunity of *Oreochromis mossambicus*, and protein hydrolysis products are responsible for this stimulation | [76] |
| Jundiá (*Rhamdia quelen*) | 55 days | Fish viscera silage produced from acid ensiling | Fish viscera silage as a high-nutritional-quality and highly digestible nutrient source for jundiá juveniles | [29] |
| Tambaqui (*Colossoma macropomum*) | 21 days | Acidified fish silage, and fermented fish silage with 5% yogurt and 15% of different carbohydrate sources (molasses, wheat bran, and cassava waste) were produced with 0.25% antifungal agent | Acidified and fermented fish viscera silages function as a energy-rich components in aquafeed due to their high fat content in dry matter, and they are efficiently digested in the diets of juvenile tambaqui; further assessment is required to determine the optimal inclusion level of viscera silages in aquafeeds | [32] |
| White shrimp (*Litopenaeus vannamei*) | 56 days | Acid-treated fish silage | Replacing fish meal with acidified fish silage at a 25% inclusion level results in superior growth performance in white shrimp | [89] |

**Table 1.** *Cont.*

| Aquatic Animal | Feeding Trial | Ensiling Conditions | Results | Reference |
|---|---|---|---|---|
| African catfish (*Clarias gariepinus*) | 14 days | Fermented shrimp head waste meal was produced by fermenting with *Lactobacillus plantarum* using carbohydrate substrates such as cane molasses | Replacing fish meal with 30% fermented shrimp head waste meal can be a cost-effective and sustainable option in the diet of African catfish | [106] |
| Mozambique tilapia (*Oreochromis mossambicus*) | 52 days | Fish silage oil recovered from fish processing waste | Fish silage oil effectively substitutes the control oil without any negative effects on production performance, while improving cellular non-specific immunity and simultaneously decreasing total mortalities; additionally, fish silage oil is a cost-effective alternative dietary oil for tilapia diets | [74] |
| South African abalone (*Haliotis midae*) | 153 days | Fish silage oil recovered from fish processing waste | Incorporating fish silage oil can enhance cellular immune function in *Haliotis midae*, but it is important to optimize the inclusion level to counteract any negative effects on production efficiency | [75] |
| Barramundi (*Lates calcarifer*) | 56 days | Fish protein hydrolysate was prepared through the fermentation of tuna fish waste using baker's yeast *Saccharomyces cerevisiae* (instant dried yeast) and *Lactobacillus casei* | Replacing fish meal with tuna protein hydrolysate at 50% and 75% inclusion levels negatively impacted the growth, feed utilization, and digestibility of juvenile barramundi | [107] |

De Arruda et al. [90] found that up to 15% acidified fish silage can be used as a partial substitute for fish meal in the formulation of carnivorous fish feed. Liang et al. [91] observed enhanced growth performance in Japanese sea bass (*Lateolabrax japonicus*) when 15% of fish meal was replaced with silage protein hydrolysate. Espe et al. [92] demonstrated that replacing less than 15% of fish meal with silage protein hydrolysate in diets for Atlantic salmon (*Salmo salar*) resulted in improved growth. However, increasing the level of silage protein hydrolysate inclusion led to a reduction in growth. In situations where diets are rich in plant-based proteins, a concentrated protein hydrolysate from fish silage can provide essential amino acids and non-amino acid nitrogen compounds that attract fish to the feed. This protein hydrolysate is particularly valuable for its supply of essential amino acids and taurine, which are typically absent in plant-based ingredients [92,108]. In the study by Plascencia-Jatomea et al. [93], shrimp head silage protein hydrolysate was identified as a promising alternative protein source for feeding tilapia, demonstrating improved growth ratios even at dietary inclusion levels as high as 15%.

The study by Ridwanudin and Sheen [94] noted that combining 10% or 20% fish silage with poultry by-product meal could replace 50% of fish meal protein in the diets for orange-spotted grouper (*Epinephelus coioides*) without any adverse effects on growth performance. Previously, Fagbenro and Jauncey [95,96] and Fagbenro and Bello-Olusoji [97] demonstrated that when co-dried with protein feedstuffs, fermented fish silage serves as a suitable protein supplement in catfish diets. It can provide up to 50–70% of dietary protein without adversely affecting feed efficiency, fish growth, or health. Several studies have shown that fermented fish silage exhibits higher digestibility (over 80%) in tilapia and catfish diets, and fish silage pellets maintain excellent physical stability, demonstrating superior water resistance [46,47,67,98]. In the study by Soltan and El-Laithy [99], fish by-products underwent a fermentation process with *Lactobacillus plantarum* at 30 °C. The liquid silage was blended with dried tomato by-product meal and potato by-product meal in a 40:30:30 ratio (*w/w/w*) and then sun-dried. The results suggest that replacing 30% of dietary

protein with fermented silage in Nile tilapia diets could be a viable option, potentially reducing feed costs without adversely affecting growth or feed utilization parameters.

Soltan and Tharwat [100] demonstrated that replacing 25% of fish meal in tilapia diets and 50% of fish meal in catfish diets with dried fermented fish silage did not significantly adversely affect the growth or feed utilization parameters of Nile tilapia and catfish. In another study, Soltan et al. [101] showed that substituting up to 50% of fish meal with fermented fish silage (composed of 60% fish waste and 30% rice bran) did not have any negative effects on the growth and feed utilization of tilapia. Additionally, this substitution resulted in a reduction in feeding costs for Nile tilapia and catfish diets [101,102].

Sun et al. [103] and Mondal et al. [104,105] reported that mixtures of fermented fish by-products and agricultural by-products could replace up to 30% of fish meal in the diets of juvenile olive flounder and catfish. Davies et al. [87] found that fish silage produced through fermentation with organic acids or LAB and carbohydrate sources, such as apple pomace or molasses, effectively served as a partial replacement for fish meal in the diet of juvenile European sea bass (*Dicentrachus labrax*). The fermentation processes applied to fish and agricultural by-products enhance their storage stability and minimize nutrient loss. The nutritive values of these by-products can be improved by the fermentation process [87,102–104].

The study by Goosen et al. [76] found that the inclusion of acidified fish viscera silage at a low level increased phagocytic activity and improved the growth performance of Mozambique tilapia (*Oreochromis mossambicus*). Fish viscera silage can serve as a valuable source of dietary protein and essential amino acids in tilapia diets [76]. Some plant-based ingredients experience deficits in amino acids, such as lysine and methionine. Thus, incorporating fish viscera silage into the mixture of plant ingredients may be sufficient to meet the nutritional needs of omnivorous fish [32]. The research conducted by Banze et al. [29] suggests that fish viscera silage can be an alternative to fish meal. The acidified fish viscera silage showed a good nutritional composition, including a high protein content, favorable amino acid profile, and essential fatty acids, as well as a good microbiological quality. Protein digestibility was similar to that of fish meal, but dry matter digestibility was higher. Overall, fish viscera silage presents high nutritional quality and digestibility for jundia juveniles, suggesting its potential as an alternative protein source in aquafeeds [29]. Furthermore, acidified and fermented fish viscera silages function as energy-rich components in aquafeed due to their high fat content in dry matter, and they are efficiently digested in the diets of juvenile tambaqui. Fish viscera silage can serve as an alternative ingredient of high biological value in fish feed production, providing nutrients and also acting as a palatability enhancer [32].

Fish silage is considered a potential substitute for fish meal in white shrimp feed. Research conducted by Shao et al. [89] indicated that replacing fish meal with acidified fish silage at a 25% inclusion level resulted in superior growth performance of white shrimp (*Litopenaeus vannamei*) compared to higher replacement levels. Furthermore, the incorporation of dietary fish silage appeared to regulate shrimp growth through the mammalian target of rapamycin (mTOR) signaling pathway [89]. On the other hand, shrimp waste can be reused by creating silage using organic or inorganic acids or fermentation technology for long-term preservation. The research found that ensiling shrimp waste with LAB stabilized carotenoids and promoted their recovery. The resulting product had antioxidant activity and contained carotenoids. It could be used as a nutrient ingredient in aquaculture feed formulations to achieve a positive effect [109]. Research conducted by Nwanna [106] reported that based on the estimated economic benefits and nutrient utilization indices, shrimp head silage meal can effectively replace fish meal up to 30% in the diet of African catfish.

Research has explored the replacement of conventional fish oil with silage oil in fish diets, revealing that silage oil is a cost-effective feed ingredient with several advantages over traditional fish oils, especially in tilapia feeds [74]. It serves as a valuable source of essential PUFAs and demonstrates antimicrobial properties within both the feed and the gastrointestinal tract of experimental fish. Moreover, the inclusion of silage oil in diets

has been shown to significantly enhance non-specific immunity at the cellular level while simultaneously reducing overall mortality rates [74,75]. The improvement in the phagocytic activity of leukocytes, which leads to enhanced cellular non-specific immunity, is likely attributed to the optimized dietary fatty acid balance present in silage oil [74,110].

Utilizing fish silage and/or fish silage oil as feed ingredients provides a sustainable and cost-effective solution, offering essential nutrients and the potential for growth-promoting effects in farmed fish. However, it is crucial to consider the inclusion level of fish silage in aquafeeds and its impact on various aspects, including growth performance, feed efficiency, nutrient utilization, carcass chemical composition, overall fish health, and gut morphology. Higher levels of replacement have been observed to reduce growth performance and can significantly affect fish health [87,89,92,100].

The study conducted by Goosen et al. [76] highlighted the presence of an upper limit for incorporating fish silage into fish diets. Elevated levels of fish silage had a detrimental effect on growth and led to increased tilapia mortality. This decline in growth was associated with the introduction of excessive free amino acids and short-chain peptides from extensively hydrolyzed fish silage, which could potentially affect taste quality and cause metabolic disruptions. Throughout the trial, experimental animals experienced heightened stress levels, resulting in elevated mortality rates [76].

In the research by Shao et al. [89], high replacement levels (75% and 100%) of fish meal with fish silage depressed the growth performance of white shrimps. Feeding juvenile barramundi with diets containing high levels (50% and 75%) of tuna protein hydrolysate resulted in decreased growth and digestibility in the fish, along with the observation of abnormal signs in liver histopathology [107]. The optimum level of tuna protein hydrolysate inclusion in aquaculture diets is species-specific, with most studies reporting inclusion levels of up to 30% [107,111]. Equally important to note is that in the case of farmed fish, feeding them with processed animal protein from the same farmed species should be avoided [27].

## 5. Fish Silage as Feed Ingredients: Advantages, Challenges, and Considerations

Fish silage, derived from fish waste or discarded fish, is increasingly recognized as a valuable ingredient in aquafeeds, offering both advantages and disadvantages that need to be addressed. Utilizing fish silage as a feed ingredient offers several advantages, including the following:

(i) Waste utilization and environmental sustainability: Ensiling repurposes fish waste and discarded fish unsuitable for human consumption or conventional fish meal production. Fish silage production can address environmental concerns by providing a proper disposal method for fish waste, mitigating the impact of inadequate waste management practices [7–10,12,27,112,113].

(ii) Cost-effectiveness: Fish silage can be a cost-effective alternative to traditional fish meal, which is often more expensive. Incorporating fish silage into aquafeeds can reduce feeding costs, minimizing overall production expenses by utilizing fish waste instead of costly feed ingredients. Several studies have shown that adding fish silage to aquafeeds can reduce feeding costs by replacing fish meal with fish silage [7,27,101,102,112,113].

(iii) Nutrient-rich feed ingredient: Fish silage is a source of hydrolyzed proteins and lipids, providing essential nutrients for animals and promoting growth and health. Fish silage has similar nutritional qualities to fish meal but offers improved digestibility due to hydrolyzed proteins and lipids. It can be a good source of protein, essential amino acids, and fatty acids, but its nutritional composition may vary depending on the raw materials used and the processing methods [7,27,29,31–34,36,37,66].

(iv) Improved feed conversion: Fish silage positively impacts feed conversion rates, leading to more efficient conversion of feed into biomass, resulting in improved production outcomes and a reduced environmental impact [27,32,37,38].

(v)  Simple and accessible process: The production of fish silage is a simple and cost-effective process compared to traditional fish meal production, making it more accessible, especially for small-scale operations [7,27,28,112,113].

Some of the key issues and limitations associated with scaling up commercial production and the use of fish silage as a feed ingredient are expected to include the following:

(i)  Natural compositional variability and quality control: The composition and quality of fish silage can vary due to factors like the type of raw materials used, their freshness and microbiological status, and the fermentation conditions and processing methods employed. This variability can make it challenging to ensure consistent quality and nutritional value in the final product [29,30,32,112,113].

(ii)  Accessibility and availability of raw materials: The availability of fish waste or discarded fish as raw materials for fish silage production may be limited or seasonal, affecting the stability of supply and the expansion of fish silage production [112,113].

(iii)  Transport and storage challenges: The high water content of fish silage poses difficulties during transportation and storage, leading to increased costs. Specialized handling and storage conditions may be required to maintain its quality. For example, it must be stored in airtight containers to prevent the ingress of oxygen, which promotes the growth of aerobic pathogens and leads to spoilage [7,27,112–115].

(iv)  Processing costs and energy consumption: Although ensiling is recognized as a cost-effective process compared to traditional fish meal and fish oil production, some fish silage production methods may require additional energy and increase the processing costs. The advanced methods, such as freeze drying, spray drying, or encapsulation, would likely increase the energy consumption and overall processing costs compared to basic fish silage production [31,114,115].

(v)  Processing time: Fish silage production can be time-consuming, involving specific steps such as acidification, liquefaction, fermentation, or drying processes, which can affect production efficiency [27,114,115].

(vi)  Microbial contamination: Fish silage, especially when not properly stored or subjected to drying or encapsulation, can be susceptible to microbial contamination, potentially affecting its safety and shelf life [27,112–114].

(vii)  Biogenic amine formation: The formation of biogenic amines is a critical issue during fermentation, as it can compromise feed safety and fish health if not properly controlled [26,29,31,33–35,59,77,78].

Some proposed considerations for improving the quality and expanding the scale of commercial production and the use of fish silage as a feed ingredient are expected to include the following:

(i)  Ensuring a stable and reliable supply of raw materials: Securing a consistent supply of fish waste while maintaining the quality of raw materials is crucial for the commercial viability and scalability of fish silage production [112,113].

(ii)  Separation and processing of different fish species: Separating certain fish species or parts of fishes may be beneficial, especially considering regulations around same-species feeding. This can help ensure a more consistent and higher-quality final product [27,29].

(iii)  Developing a coordinated collection and transport network: To expand commercial production, a well-organized system for collecting and transporting the fish waste and discarded fish to the processing place is needed, maintaining proper temperature conditions to maintain the quality of raw materials [7,112,113].

(iv)  Advanced processing techniques: Utilizing advanced processing techniques such as spray drying, encapsulation, and refractance window drying can enhance the quality control and preservation of the final product. These methods can address challenges like high water content, which can complicate the transportation and storage of fish silage, despite the associated increase in production costs. On the other hand, co-

drying the fish silage with other ingredients like soybean, corn, barley, or wheat bran can produce a more stable and easier to handle product [31,34,114–119].

(v) Optimization of the fermentation process: Formation of biogenic amines is a critical issue during fermentation. To reduce the risk of biogenic amine formation, it is necessary to monitor acceptable levels of biogenic amines in fermented feeds, optimize fermentation conditions (such as time, temperature, moisture content, and salt concentrations), and select suitable strains of lactic acid bacteria that do not produce biogenic amines. The optimization of the fermentation process is important for controlling the formation of biogenic amines and ensuring the safety and quality of the final fermented product [26,29,31,33–35,59,60,77,78].

(vi) Addressing regulatory and legislative requirements: In developed markets, fish silage production must comply with strict quality standards and legislation, which may require additional processing steps [112,113].

These considerations highlight the multifaceted approach required to improve the quality and expand the commercial production of fish silage.

## 6. Emerging Technologies for Enhancing the Nutritional Value and Efficiency of Fish Silage Production

Fish silage, typically produced and stored in a liquid state, faces challenges such as transportation and storage due to its high water content, leading to increased costs [7,27,31]. The elevated water content in fish silage also limits its direct use in dry or moist feeds. Additionally, it needs to be stored in airtight containers to prevent the ingress of oxygen, which can promote the growth of aerobic pathogens and lead to spoilage [114].

Drying silage offers numerous advantages, including increased value, ease of handling, reduced packaging, transportation, and storage costs, microbial growth control, and higher protein concentration for diets requiring increased protein content. However, traditional drying methods are expensive and environmentally harmful due to high energy consumption. One environmentally friendly alternative approach is solar drying. Despite its economic efficiency, solar drying has drawbacks such as longer processing times, uneven drying, and the possibility of microbial contamination [114].

### 6.1. Spray Drying Encapsulation

A noteworthy alternative gaining significant attention is encapsulation through spray drying. Spray drying is a commonly applied technology for reducing moisture in liquid materials and transforming them into powdered products, offering various advantages [31,120–124]. The liquid or slurry is sprayed into a hot gas stream, where the water or solvent evaporates, leaving behind the dried powder. This method is particularly suitable for heat-sensitive materials, providing benefits such as ease of use, short drying times, cost-effectiveness, and the production of high-quality microcapsules [31,120–124]. Encapsulation involves enclosing or entrapping the active ingredient or substance within a protective or carrier material, forming capsules or particles. This is often performed to protect the active ingredient from degradation or to control its release. The spray drying approach for producing fish silage can effectively preserve the bioactive properties and nutritional value of the final product, making it a suitable feed component. Özyurt et al. [31] reported that spray dried fish silages have substantial potential as feed components due to their high digestibility rate and nutritional components, as confirmed by nutritional and chemical evaluations.

The process involves utilizing high temperatures that do not exceed the wet bulb temperature range of 30–50 °C, thereby preventing damage to the product. Spray drying encapsulation entails coating the core material with a carrier agent, which plays a crucial role in preserving the bioactive properties of the product while concealing its sensory attributes [123,124]. Maltodextrin is a commonly used wall material in the industry due to its affordability, neutral taste and aroma, and low viscosity at high solid concentrations. It acts as a robust barrier against core material oxidation and external factors. Microen-

capsulation using carrier agents like maltodextrin offers significant industrial advantages and facilitates the development of innovative functional foods [115]. However, the low emulsifying capacity of maltodextrin is a disadvantage. Therefore, some researchers have suggested combining maltodextrin with other surface-active biopolymers such as gum arabic, modified starches, and proteins to provide more effective microencapsulation by spray drying [115,124].

Spray drying can also be employed as an effective method for microencapsulation of fish oils, particularly suitable for heat-sensitive materials like omega-3 fatty acids and phytosterols. This process offers numerous benefits, including safeguarding the encapsulated substances from undesirable reactions like lipid oxidation and nutritional degradation during various stages of production, handling, and storage. Moreover, microencapsulation helps in retaining volatile components and masking undesirable tastes [120,122,123,125]. However, for maximum protection during processing and storage, it is often necessary to include antioxidants. Citrus essential oils, derived from citrus peels rich in flavonoids and phenolic compounds, have gained attention for their potential biological properties, including antimicrobial, antioxidant, anticancer, and anti-inflammatory effects [122]. These essential oils can be used as natural preservatives and additives in food and beverage products. Microencapsulation plays a crucial role in protecting PUFAs from oxidation by creating a physical barrier between the active material and the environment. Various factors influence the oxidative stability of encapsulated oils, such as the choice of wall materials, antioxidants, oil quality, particle characteristics, and storage conditions [122].

### 6.2. Microencapsulation of Bioactive Compounds

Microencapsulation is a technology that is finding increasingly wide application in animal nutrition. This technology allows for the protection of sensitive compounds during delivery and storage conditions. Additionally, the use of this method makes it possible to increase the bioavailability of compounds and mask the unpleasant taste of compounds [126]. Encapsulation can be used to enhance the viability and efficacy of LAB in feed production. Encapsulation helps protect the LAB from harsh environmental conditions, such as low pH, during the feed production process. *Weissella paramesenteroides*, a LAB strain, has been evaluated for encapsulation as a technological tool for use in biological fish silage production [127]. The encapsulation of *W. paramesenteroides* using an alginate method achieved an encapsulation efficacy of 85%, indicating it is an effective method to enhance the viability of *W. paremesenteroides* and could be a reliable alternative to ferment fish waste. The encapsulated *W. paramesenteroides* showed no significant difference in pH reduction compared to the non-encapsulated strain during fish silage fermentation, but the encapsulation method helped maintain the bacterial viability over time, which is an advantage for technological application. The encapsulation of LAB strains like *W. paramesenteroides* can be an effective way to improve their viability and efficacy in fish silage production, helping to control spoilage and pathogenic microorganisms during the fermentation and storage of the silage [127]. Encapsulation is a crucial technology for enhancing the viability and efficacy of LAB and other sensitive feed additives, contributing to improved animal nutrition and performance.

### 6.3. Refractance Window Drying Technology

Refractance window (RW) drying is a nonthermal dehydration technique that offers several advantages over traditional drying methods. It uses heat transfer modes (conduction, convection, and radiation) and the principle of light refraction to dehydrate products, avoiding overheating and making it suitable for heat-sensitive products. This technology allows for enhanced drying rates with short processing times (2–6 min) and low product temperatures. The design of RW dryers is simple and unique, promising cost-effectiveness, energy efficiency, and ease of operation and maintenance. RW drying can help retain the color, flavor, aroma, and nutritional content of dried products, often proving comparable or superior to freeze drying [116,117].

RW drying technology applies to drying products by conveying them in a thin layer on a polyester film over a warm water bath, during which conductive, convective, and radiative heat transfer occurs. Recently, several studies have introduced the use of RW drying for drying fruit and vegetable products. Results show that RW drying can efficiently dry products under gentle conditions with relatively short processing times, while preserving product quality. Additionally, improved energy efficiency and lower costs, compared to freeze drying and spray drying, are associated with RW drying technology [116–118].

van 't Land et al. [119] investigated changes in the nutritional quality of fish silage during RW drying. Furthermore, the effect of the protein hydrolysis degree of fish silage on production efficiency (moisture evaporated and production rate) was determined. To obtain a dried product, fish silage was conveyed at a speed of 30 cm/min on a polyester film over a water bath at 55 °C. The moisture content decreased with a slight increase in TBARS (thiobarbituric acid reactive substances). A higher degree of protein hydrolysis in fish silage increased evaporation efficiency but reduced the production rate. The end-product consistency seemed to be affected by the application thickness, ambient conditions, and the degree of protein hydrolysis. The authors concluded that RW drying appears to be an effective method for rapidly drying fish silage under gentle conditions [119].

## 7. Innovative Approaches to Sustainable Protein Alternatives through Waste Valorization

The aquaculture industry is actively seeking innovative, cost-effective, and sustainable protein alternatives to traditional fish meal and plant-based ingredients. There is currently significant interest in utilizing by-products from agricultural waste streams, such as fruit and vegetable biomass, in aquaculture applications. This approach holds considerable potential as a sustainable protein source for aquafeeds [8,87]. Repurposing agricultural by-products as protein sources can alleviate the strain on land-based feed production and waste generation, promoting a closed-loop economy and fostering sustainable, eco-friendly aquaculture practices. The utilization of fish waste and agri-food waste through ensiling technology in animal nutrition as an alternative feed ingredient is represented in Figure 2.

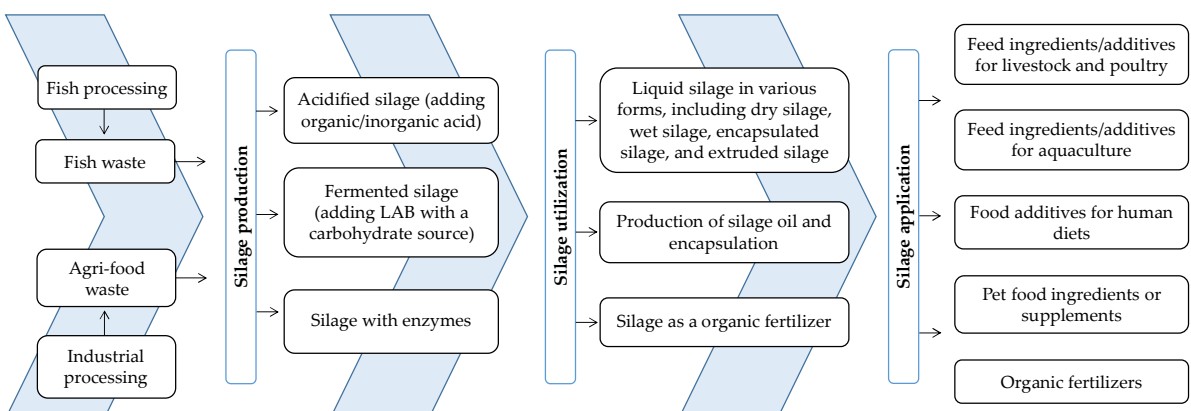

**Figure 2.** Valorization of waste to create sustainable feed alternatives through ensilaging technology.

The fermentation of agri-food waste presents a promising avenue for sustainable development by converting these waste materials into valuable, value-added products, including animal feed. Fermentation has been recognized as a convenient, environmentally friendly, and cost-effective technique, specifically in addressing challenges related to poor digestibility and the potential for cross-contamination by hazardous microorganisms in waste materials [66,87,128–131]. Tropea et al. [66] introduced a method of food waste valorization that involves bioconversion into animal feed through a fermentation process. This method repurposes non-sterilized fish waste, supplemented with lemon peel to enhance filler content and introduce prebiotics. The process combines starter cultures of *Saccharomyces cerevisiae* and *Lactobacillus reuteri*, resulting in a high-protein supplement enriched

with beneficial microorganisms suitable for aquaculture feeds. The fermented product exhibits low levels of spoilage microorganisms and a strong presence of healthy beneficial microorganisms, providing sufficient protein and lipid content to address the scarcity of protein sources in aquaculture. This innovative method encourages the conversion of fish waste and lemon peel into valuable additional feed components [66].

Furthermore, there is a growing interest in utilizing agricultural waste by-products, such as apple pomace from cider production, which is rich in fermentable sugars and pectin. Combining apple pomace with marine biomass to produce fish silage for aquafeed offers an opportunity to add value and align with sustainable raw material utilization. This concept could potentially be extended to utilize other waste sugar sources from fruit and vegetable processing, thus mitigating environmental impacts and creating additional protein sources for aquaculture [87,132].

Munekata et al. [133] conducted a comprehensive review on the utilization of pomaces, which are waste products generated during the extraction of juices and olive oil from fruits and olives. The study thoroughly investigated the valorization of pomaces and their incorporation into animal feed. The production of silages and feeds from fermented pomaces can enhance animal health and represents a viable alternative. This approach acknowledges the potential impact on growth performance while emphasizing the improvement in animal health status. Furthermore, it highlights the absence of negative effects and the enhancement of the nutritional quality of food derived from animals fed with fermented pomaces, underscoring these as favorable characteristics that support this strategy [133].

The study conducted by Panyawoot et al. [134] evaluated the effects of feed obtained through fermentation on the final consumers. Their study examined the impact of fermented discarded durian peel, a seasonal fruit widely grown in tropical countries, in the diets of growing crossbreed Thai Native–Anglo-Nubian goats. The durian peel was subjected to fermentation using a combination of molasses, *Lactobacillus casei*, and cellulase. Different treatments, including separate and combined applications of these components in total mixed rations, were assessed for their effects on various aspects such as feed utilization, digestibility, ruminal fermentation, and nitrogen utilization in the goats' diet. The study revealed that when the discarded durian peel was fermented with a combination of molasses and *L. casei*, it led to significantly higher nutrient digestibility and propionate concentration. Moreover, there were observed decreases in estimated methane production, the acetate-to-propionate ratio, and urinary nitrogen compared to untreated discarded durian peel. Therefore, using fermented durian peel with *L. casei* could contribute 25% of dry matter to the diet of growing goats without a negative impact [134].

## 8. Utilization of Fish Silage as a Fertilizer

Fish silage can be used as a fertilizer if it does not meet the quality standards for use as animal feed [27,48,49,135]. It is a valuable organic fertilizer due to its nutrient-rich composition, containing nitrogen, phosphorus, potassium, calcium, magnesium, and trace elements that may not be present in some industrial fertilizers needed by plants. The nutrient composition of the fish silage can vary depending on the raw materials used, with a higher bone content resulting in increased levels of elements like phosphorus and magnesium [27,48,49]. Free amino acids and short peptides in fish silage can be directly absorbed by plants, stimulating growth. Amino acids play a crucial role in plant growth and cellular processes, including pH regulation and defense against environmental stresses. It has been reported that fish silage, obtained from fish waste, is rich in L-amino acids, especially free amino acids, which could be a cost-effective and excellent alternative as a plant growth enhancer or organic fertilizer to increase crop yields [48,49,136].

Fish silage, when used as a fertilizer, can be effectively applied by adding approximately 2–5% of liquid silage to irrigation water [27]. It has been reported that crop yields obtained using fish silage as a fertilizer have been comparable to those obtained using traditional fertilizer. Karim et al. [48] investigated the physicochemical properties of liquid fish silage and its effects on the growth, yield, pigment content, and post-harvest quality

of pak choi (*Brassica rapa* L. subsp. *chinensis*). Fish silage treatments were prepared at five concentrations (1%, 2.5%, 5%, 7.5%, and 10%) and compared with plants fertilized with commercial fertilizer (N-P-K 15:15:15) at recommended nutrient levels per hectare. After 14 days of fermentation, the fish silage contained 1.84% N, 0.50% P, 0.41% K, 0.36% Mg, and 0.84% Ca. The results showed that fish silage at 5%, 7.5%, and 10% produced similar growth, yield, pigment content, and post-harvest quality as plants fertilized with commercial fertilizer. The study authors recommend using 5% fish silage as it is more economical compared to the 7.5% and 10% concentrations [48].

Owing to their low molecular weight and short peptide length, free amino acids from fish waste can be easily assimilated by the roots and stromata of plants. The study by Gauthankar et al. [49] compared the biochemical and nutritive properties, particularly the amino acid composition, of acidic silages from fatty Indian mackerel and lean false trevally over a 35-day fermentation period. Indian mackerel silage had a higher total and free amino acid content, with specific amino acids like asparagine, histidine, isoleucine, valine, cysteine, serine, lysine, and arginine being more abundant. Both silages contained essential amino acids such as leucine, glutamic acid, and arginine and showed significant changes in composition with fermentation duration. The study suggests that a 25–30-day fermentation period significantly affects amino acid composition, making fish silage a potentially valuable organic fertilizer [49].

Fish silage, with its nutrient-rich composition, offers a viable alternative to conventional fertilizers, potentially reducing environmental waste and pollution. Its affordability, safety, and nutritional value also make it a promising plant growth promoter. Further research is required to elucidate the impact of fish species and processing techniques on its nutritional content. This approach resonates with the principles of the circular economy and organic farming, as it seeks to repurpose fish waste and discarded fish into organic fertilizers. This initiative promotes the responsible recycling of nutrients originally sourced from marine ecosystems and reintroduces them into terrestrial landscapes, contributing to sustainable agriculture practices [48,49,135].

## 9. Conclusions and Future Remarks

Sustainability in the aquaculture industry hinges on finding alternative protein sources to replace fish meal, aligning with the principles of Blue Growth that emphasize sustainable marine resource development and waste reduction. Repurposing fish waste and discarded fish, often regarded as waste and pollution sources, offers an opportunity to enhance sustainability by using them in animal and aquafeed production, as well as for organic fertilizer. The use of organic acids, such as formic acid, to preserve fresh fish raw materials is a cost-effective and scalable technology that transforms waste into valuable products, contributing to sustainability efforts. Fish silage and protein hydrolysate derived from it are valuable ingredients in animal and fish feed when used appropriately. Additionally, the organic acids used in the production of fish silage have antimicrobial and antioxidant properties, enhancing the safety and preservation of the feed. Incorporating fish silage into fish feeds can improve feed acceptance, stimulate non-specific immunity, and enhance growth rates in fish. Although the incorporation of formic acid and short-chain organic acids in aquaculture feed holds promise, further research and optimization are necessary to maximize their benefits. Combining agricultural food waste from fruit and vegetable processing with marine biomass to produce fish silage for aquafeed presents an excellent opportunity to add value, align with sustainable raw material utilization, mitigate environmental impacts, and create additional protein sources for aquaculture.

Looking ahead, the incorporation of fish silage represents an excellent opportunity for protein replenishment, and its inclusion in aquafeeds holds significant potential for enhancing nutritional quality and overall sustainability in aquaculture. Fermentation processes not only enhance nutrient availability and digestibility but also promote the presence of beneficial microorganisms, positively impacting the health and growth of aquatic species. Furthermore, utilizing fish silage ingredients contributes to reducing

the environmental footprint of aquaculture by repurposing food waste and by-products, aligning with the circular economy principles in the industry. This direction holds great promise for the future of sustainable aquaculture practices.

Unlocking the full benefits of fermented feed ingredients for aquaculture requires further research and development. Advancements in fermentation techniques, microbial culture optimization, and comprehensive nutritional assessments will be crucial in maximizing the positive effects on aquafeed formulations. Additionally, investigations into the long-term effects of fermented feed on aquatic organisms, including potential impacts on gut health, disease resistance, and product quality, remain areas of interest. As the aquaculture industry continues to evolve towards more sustainable and efficient practices, the exploration of fermented feed ingredients is poised to play a pivotal role in shaping its future.

**Author Contributions:** Writing—original draft preparation, A.M.; writing—review and editing, A.M.; resources, A.M. and L.B.; supervision, A.P. and L.T.; project administration, O.S. All authors have read and agreed to the published version of the manuscript.

**Funding:** This research received no external funding.

**Acknowledgments:** We would like to express our gratitude to the referees for their thoughtful comments and recommendations.

**Conflicts of Interest:** The authors declare no conflicts of interest.

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
