# Peer review of "Exploring Sustainable Aquafeed Alternatives with a Specific Focus on the Ensilaging Technology of Fish Waste"

_fermentation, doi:10.3390/fermentation10050258_

Round 1
Reviewer 1 Report
Comments and Suggestions for Authors
The Abstract should simplify.
Line 23 Silage is not a feed additive.
Line 25 The key words should have no more than five.
Line 32,36 The statistical data can be updated since it was too old.
Line 60-64 Actually, a lot of soybean meal were used in aquatic feed. Please revise.
In the Introduction, please simplify the contents and provide more information related to silage.
Line 166-180 Here the author can provide a figure to show the fermentation of silage.
Line 258 In this part, the author can write separately. The addition of microorganisms and nutritive additive such as molasses can be written independently.
Line 637 Please provide more information and cite references.
The effects of fish silage on disease should pay more attention.
Did the authors conduct related experiment in fish silage? The writing of review should establish on previous researches which mainly come from your group since the foundation of research was very important. In the reference, I did not find any paper from the authors.
Author Response
Response to Reviewer 1 Comments
Thank you very much for taking the time to review this manuscript. Please find the detailed responses below and the corresponding revisions/corrections highlighted/in track changes in the re-submitted files.
1. The Abstract should simplify -
Abstract: The global increase in population has placed significant pressure on food security, leading to the emergence of aquaculture as a vital source of aquatic foods. However, rising costs and limited fishmeal availability in aquafeeds have driven the search for alternative protein sources. While plant-based ingredients have been integrated into commercial aquafeeds, they come with challenges such as low protein content, palatability issues, and the presence of antinutritional factors. In this context, fish silage, made from fish waste and discarded fish, stands out as a promising alternative technology due to its cost-effectiveness and sustainability attributes. The production of fish silage involves the addition of organic/inorganic acids or lactic acid bacteria to homogenized fish waste, yielding a valuable mixture rich in peptides and free amino acids, offering significant nutritional benefits for animal diets. This review aims to promote sustainable practices in the aquaculture industry by analyzing research results related to silage technology, appraising the advantages and disadvantages of using fish silage as a feed ingredient, and focusing on emerging trends in this field.
2. Line 23 Silage is not a feed additive - corrected to feed ingredient.
3. Line 25 The key words should have no more than five - corrected to -
Keywords: ensiling technology; fish waste; lactic acid bacteria; fish silage; waste valorization
4. Line 32,36 The statistical data can be updated since it was too old - this statistical data from this source - The State of World Fisheries and Aquaculture 2022. Towards Blue Transformation; Food and Agriculture Organization of the United Nations: Rome, Italy, 2022. I cant find a more recent source.
5. Line 60-64 Actually, a lot of soybean meal were used in aquatic feed. Please revise - Plant-based raw materials have undergone thorough investigation and are successfully included in commercial aquaculture feeds. However, utilizing conventional plant-based protein in aquaculture, particularly for carnivorous species, faces challenges including inadequate protein content, palatability issues, unbalanced amino acid profiles, and an-ti-nutritional factors.
6. In the Introduction, please simplify the contents and provide more information related to silage - I would like to keep the contents in the introduction because I want to mention other available sources of protein and promising recent technologies, such as the bioconversion of waste. I briefly discussed the ensiling technology in the introduction, as the following sections contain a more detailed description of ensiling.
7. Line 166-180 Here the author can provide a figure to show the fermentation of silage - Figure in progress; I'm working on it now.
8. Line 258 In this part, the author can write separately. The addition of microorganisms and nutritive additive such as molasses can be written independently - I don't understand this question. Could you please paraphrase?
9. Line 637 Please provide more information and cite references - added information.
10. The effects of fish silage on disease should pay more attention.
11. Did the authors conduct related experiment in fish silage? The writing of review should establish on previous researches which mainly come from your group since the foundation of research was very important. In the reference, I did not find any paper from the authors -
From April 2022 to September 2023, experiments on acid ensiling technology were conducted by the student (Magister) Leonid Belyi from the Institute of Biotechnology, Bioengineering, and Food Systems, Advanced Engineering School, Far Eastern Federal University, under my supervision. The results of these experiments were presented in a dissertation. Additionally, experiments on ensiling with LAB and enzymes have been ongoing since January 2024, under my supervision. Experimental articles are currently being prepared.

Reviewer 2 Report
Comments and Suggestions for Authors
L422 Utilization of Fish Silage in Animal Nutrition. This title needs to be revised. It gives the impression that it includes the nutrition of all animals, both aquatic and terrestrial, but the author's description below focuses mainly on aquatic animals.
L497 Section 3 had only one subtitle. Therefore, it is suggested that the Section 3 should be reorganized into 2-3 subheadings based on the context.
Comments on the Quality of English LanguageNo.
Author Response
Response to Reviewer 2 Comments
Thank you very much for taking the time to review this manuscript. Please find the detailed responses below and the corresponding revisions/corrections highlighted/in track changes in the re-submitted files.
1. L422 Utilization of Fish Silage in Animal Nutrition. This title needs to be revised. It gives the impression that it includes the nutrition of all animals, both aquatic and terrestrial, but the author's description below focuses mainly on aquatic animals.
I have revised the title to 4. Utilization of Fish Silage in Aquafeeds.
2. L497 Section 3 had only one subtitle. Therefore, it is suggested that the Section 3 should be reorganized into 2-3 subheadings based on the context.
I have renamed section 3 to 4 and given it the title '4. Utilization of Fish Silage in Aquaculture Feeds.' I have also added a new section and given it the title '3. Nutritional and Health Benefits of Fish Silage.

Reviewer 3 Report
Comments and Suggestions for Authors
The document titled "Exploring Sustainable Aquafeed Alternatives with a Specific Focus on the Ensilaging Technology of Fish Waste" presents a comprehensive review on the utilization of fish silage derived from fish waste as a sustainable and cost-effective alternative to traditional aquafeed ingredients. The review highlights the technological processes involved in producing fish silage, including acidic and fermented fish silage production, and explores its applications in animal nutrition, particularly in aquafeeds. It also discusses the environmental benefits, challenges associated with fish silage production, and the potential for fish silage to be used as a fertilizer.
- The review could benefit from a more detailed discussion on the comparative nutritional value of fish silage versus other alternative protein sources, including specific data on growth performance, feed conversion rates, and health impacts on various aquaculture species.
- There is a lack of detailed economic analysis comparing the cost-effectiveness of producing and utilizing fish silage in aquafeeds versus traditional feed ingredients.
- While the document touches on innovative approaches and future prospects, it could further elaborate on emerging technologies and research areas that could enhance the utility and production efficiency of fish silage, such as genetic engineering of microbes for improved fermentation processes or the development of new encapsulation technologies for better storage and transport.
- How does the nutritional content and digestibility of fish silage compare to that of traditional fishmeal and plant-based alternative protein sources in aquafeeds?
- What are the specific challenges and limitations associated with scaling up fish silage production for commercial aquafeed applications?
- Can the authors provide insights into ongoing research or future innovations that may further improve the sustainability, nutritional value, or cost-effectiveness of fish silage as an aquafeed ingredient?
The manuscript is well-written and adheres to scientific conventions for the most part. However, ensuring consistency in terminology and formatting according to the journal's guidelines, including the proper use of English and adherence to scientific standards for citing references and presenting data, would further improve the manuscript's quality.
Author Response
Response to Reviewer 3 Comments
Thank you very much for taking the time to review this manuscript. Please find the detailed responses below and the corresponding revisions/corrections highlighted/in track changes in the re-submitted files.
- How does the nutritional content and digestibility of fish silage compare to that of traditional fishmeal and plant-based alternative protein sources in aquafeeds? I describeb this in Chapter 3. Nutritional and Health Benefits of Fish Silage
- What are the specific challenges and limitations associated with scaling up fish silage production for commercial aquafeed applications? - I described this in Chapter 5. Utilizing Fish Silage as Feed Ingredients: Advantages, Challenges, and Considerations
- Can the authors provide insights into ongoing research or future innovations that may further improve the sustainability, nutritional value, or cost-effectiveness of fish silage as an aquafeed ingredient? I described this in Chapter 6. Emerging Technologies for Enhancing the Nutritional Value and Efficiency of Fish Silage Production

Round 2
Reviewer 1 Report
Comments and Suggestions for Authors
None
Reviewer 3 Report
Comments and Suggestions for Authors
Good.